# Inhaled Corticosteroids and the Risk of Nontuberculous Mycobacterial Pulmonary Disease in Chronic Obstructive Pulmonary Disease: Findings from a Nationwide Population-Based Study

**DOI:** 10.3390/jpm13071088

**Published:** 2023-06-30

**Authors:** Iseul Yu, Se Hwa Hong, Min-Seok Chang, Seok Jeong Lee, Suk Joong Yong, Won-Yeon Lee, Sang-Ha Kim, Ji-Ho Lee

**Affiliations:** 1Department of Internal Medicine, Yonsei University Wonju College of Medicine, Wonju 26426, Republic of Korea; dbdltmf13@yonsei.ac.kr (I.Y.); mim8776@yonsei.ac.kr (M.-S.C.); leeseokj@yonsei.ac.kr (S.J.L.); sjyong@yonsei.ac.kr (S.J.Y.); wonylee@yonsei.ac.kr (W.-Y.L.); sanghakim@yonsei.ac.kr (S.-H.K.); 2Department of Biostatistics, Wonju College of Medicine, Yonsei University, Wonju 26426, Republic of Korea; hhdtpghk@yonsei.ac.kr

**Keywords:** nontuberculous mycobacterial pulmonary disease, inhaled corticosteroid, chronic obstructive pulmonary disease, fluticasone, budesonide

## Abstract

Studies have shown increased nontuberculous mycobacterial pulmonary disease (NTM) incidence with inhaled corticosteroid (ICS) use in patients with chronic respiratory diseases; however, this association in chronic obstructive pulmonary disease (COPD) remains insufficiently studied. Using a nationwide population-based database of the Korean National Health Insurance Service, newly diagnosed COPD patients (2005–2018) treated with inhaled bronchodilators were selected. An NTM case was defined by the presence of the first diagnostic code following inhaled bronchodilator use. Results indicated that ICS users did not have an increased risk of NTM disease compared to non-ICS users (hazard ratio (HR), 1.121; 95% confidence interval (CI), 0.950–1.323; *p* = 0.176). However, in a subgroup analysis, the highest quartile of the cumulative ICS dose was associated with the development of NTM (1.200, 0.950–1.323, *p* = 0.050). Medium (1.229, 1.008–1.499, *p* = 0.041) and high daily doses of ICS (1.637, 1.241–2.160, *p* < 0.001) were associated with an increased risk of NTM disease. There was no difference in the risk of NTM according to ICS type. ICS use may increase the risk of developing NTM disease in patients with COPD. Physicians should weigh the potential benefits and risks of ICS, especially when using high doses and prolonged durations.

## 1. Introduction

Chronic obstructive pulmonary disease (COPD) is the most common type of respiratory disease. The global prevalence of COPD is estimated to be 11.7%, and over three million deaths are attributable to COPD annually [1,2]. Additionally, the prevalence and mortality rates of COPD are expected to increase over the coming decades. Inhaled medications are the mainstay of treatment for COPD. Stable patients with COPD are generally managed with long-acting bronchodilators such as long-acting muscarinic antagonists (LAMAs), long-acting β_2_-agonists (LABAs), and inhaled corticosteroids (ICSs). LAMAs are preferred over LABAs because of their superior effect in terms of preventing exacerbations and improving lung function [3]. The LABA/LAMA combination is the next treatment step for patients with recurrent exacerbation or progressive dyspnea [4].

Recent studies have shown the mortality benefit of triple inhaler combinations containing ICS compared to dual combinations (LABA/LAMA) [5]. In a real-world study, triple inhalers were more effective than dual combinations in preventing COPD exacerbation [6]. However, ICS is not recommended as a monotherapy for COPD. Instead, the treatment effect of ICS can be predicted in patients with a specific treatable trait, such as blood eosinophilia and frequent exacerbations [7]. In addition, ICS treatment is not supported in patients with a high risk of pneumonia. Pulmonary tuberculosis (TB) is a respiratory side effect of ICS and has been demonstrated in cohort and case-control studies [8]. The prevalence of nontuberculous mycobacterial pulmonary disease (NTM) has rapidly increased worldwide in recent decades [9]. Furthermore, a population-based study reported poorer prognosis and higher mortality from NTM than from TB [10]. Several studies have shown an increased incidence of NTM with ICS use in patients with chronic respiratory diseases, including asthma, COPD, and asthma–COPD overlap [11,12]. However, the association between ICS use and NTM development has not been adequately studied.

This study aimed to examine the risk of NTM disease from ICS use in patients with COPD using a nationwide, population-based cohort.

## 2. Materials and Methods

### 2.1. Data Source

All individuals living in South Korea must register with the National Health Insurance Service (NHIS). The insured individuals contribute to NHIS through monthly insurance payments based on their socioeconomic status and receive medical services from healthcare providers. As a single insurer, the NHIS offers payments to healthcare providers based on their claims. The NHIS has established and operates the National Health Information Database, a public database that includes healthcare utilization, socioeconomic and demographic information, as well as mortality data for the entire population of South Korea. The healthcare utilization database includes records of inpatient and outpatient usage (diagnosis, length of stay, treatment costs, and services received) and prescription records (drug code, days prescribed, and daily dosage). All records have been converted to corresponding electronic codes in the database and utilized for medical research purposes or to develop public health policies [13].

This study was approved by the Institutional Review Board of Wonju Severance Christian Hospital (CR319322) and adhered to the principles of the Declaration of Helsinki. As this was a retrospective study using anonymous claims data, the requirement for informed consent was waived.

### 2.2. Selection of Study Participants

The diagnostic codes were classified according to the International Classification of Diseases 10 (ICD-10). COPD cases were defined according to the corresponding diagnostic codes J42–J44 (except J430), the procedure of pulmonary function tests, and prescription of inhaled medications [14]. Data from individuals who were diagnosed with COPD at least twice between January 2005 and December 2018 were initially collected from the entire Korean population, which consists of over 50 million individuals registered in the NHIS. Those who had a diagnostic code for lung cancer at any time and did not undergo a lung function test 1 year before or after COPD diagnosis were excluded. Subsequently, half of the participants were selected using random sampling and invited for research purposes, in accordance with the national policy applied to the usage of public databases. The other exclusion criteria were as follows: diagnostic history of COPD before 2005, patient age < 40 years, no record of a prescription for the management of COPD, treatment with only oral medications without a prescription for any inhalers, prescription of different types of ICSs simultaneously, and death before the prescription of inhalers.

### 2.3. Study Design

This was a retrospective cohort study. Among the study participants, ICS users were treated with various commercial ICS products containing fluticasone propionate, budesonide, beclomethasone, ciclesonide, and fluticasone furoate for at least 1 month, regardless of the use of metered-dose or dry powder inhalers. Non-ICS users were defined as individuals who received treatment with bronchodilators, including short-acting β_2_-agonist (SABA), LABA, LAMA, and the LABA/LAMA combination, without any recorded prescriptions for ICS. The index date for ICS users was defined as the first prescription date for ICS, while for non-ICS users, it was defined as the first date of any prescribed inhaled bronchodilator.

Comorbidities were diagnosed based on the corresponding diagnostic codes for each disease. The comorbidities included in this study were bronchiectasis (J47), diabetes (E10–E14), hypertension (I10), heart failure (I11 and I50), stroke (I60–I69), chronic kidney disease (N17–N19), and chronic liver disease (K70–K76). Charlson comorbidity indices (CCIs) were calculated using diagnostic codes. Bronchodilators were classified according to prescription records up to one year after the index date. Hospitalization with a COPD diagnosis 1 year before the index date was collected. Prescription records of oral corticosteroids (OCSs) were collected to determine the proportion of subjects who had received an OCS prescription and to calculate the mean duration of the OCS prescription for a period of one year before the index date [15]. These two factors were used to determine baseline COPD severity.

### 2.4. Definition of Outcomes

An NTM case was defined as the first diagnostic code (A31) after the index date [9,10]. Further analysis was conducted to determine whether NTM prevalence differed according to cumulative ICS dose, daily ICS dose, and ICS types. All ICS types were converted into equivalent doses of fluticasone propionate. Fluticasone propionate (50 μg) was equivalent to beclomethasone (100 μg), beclomethasone (50 μg), budesonide (80 μg), ciclesonide (32 μg), and fluticasone furoate (10 μg) [16]. The cumulative ICS dose was the sum of all prescribed ICSs for the entire study period and divided into quartiles. The daily dose of ICS was calculated by dividing the total prescribed ICS doses by the duration of the prescription. These doses were then classified into three categories based on fluticasone propionate as a reference: low (1–499 μg), medium (500–999 μg), and high doses (≥1000 μg).

### 2.5. Statistical Analysis

To reduce selection bias, ICS and non-ICS users were subjected to propensity score matching in a 1:1 ratio. Matching between the two groups was performed using logistic regression, incorporating various variables such as age, sex, comorbidities, hospitalization, OCS prescription rates and durations, and intervals from COPD diagnosis to the index dates. Baseline clinical variables of ICS users and non-ICS users were compared using Student’s *t*-test for continuous variables and the chi-square test for categorical variables. Cox proportional hazards analyses were conducted to analyze the hazard ratio (HR) and corresponding 95% confidence interval (CI). These analyses aimed to evaluate the relative risk of developing NTM disease associated with ICS use. Significant differences in the baseline characteristics between the two groups after propensity score matching were included in the multivariate analysis. All statistical analyses were performed using SAS 9.4 (version 9.4; SAS Institute Inc., Cary, NC, USA). Statistical significance was set at *p* < 0.05.

## 3. Results

### 3.1. Demographics

The total number of patients diagnosed with COPD without lung cancer between 2005 and 2018 and who underwent pulmonary function tests before or 1 year after COPD diagnosis was 1,545,246. After applying the aforementioned 50% sampling method and exclusion criteria, a total of 156,475 patients were included in the study. According to our definition, there were 108,294 individuals classified as ICS users and 48,181 individuals classified as non-ICS users. Propensity score matching was performed at a 1:1 ratio. Finally, an equal number of 46,197 patients was selected from the ICS and non-ICS users (Figure 1).

The mean age was 66.59 ± 10.94 years in ICS users and 66.68 ± 10.89 years in non-ICS users. The percentage of male participants was 72.82% among ICS users and 73.44% among non-ICS users (*p* = 0.034). SABA, LAMA, and LABA/LAMA combinations were prescribed more frequently to non-ICS users compared to ICS users; however, LABA was more commonly used among ICS users than non-ICS users (all *p* < 0.001). The proportion of hospitalizations due to COPD before the index year was 6.39% for ICS users and 5.25% for non-ICS users (*p* < 0.001). The prescription rates of OCS were 75.25% and 74.50% for ICS and non-ICS users, respectively (*p* = 0.008). The interval from COPD diagnosis to the index date was shorter for ICS users (*p* < 0.001). No differences were observed in the distribution of comorbidities (Table 1).

### 3.2. Incidence of NTM in ICS Users

When the daily ICS doses were classified as low, medium, or high, they accounted for 59.04%, 33.34%, and 7.61% of the total, respectively (Table 2). The most frequently prescribed ICS was fluticasone propionate (46.97%), followed by budesonide (22.08%), fluticasone furoate (16.60%), beclomethasone (10.26%), and ciclesonide (4.09%). There was no significant difference in the rate of NTM disease after the index date between the group of 495 ICS users and the group of 440 non-ICS users (*p* = 0.071). The incidence of NTM per 100,000 person-years was 228.93 for non-ICS users and 241.08 for ICS users. There was an observed increasing trend of NTM disease associated with higher cumulative doses of ICS. The incidence rates of NTM according to the cumulative dose of ICS were 188.42, 215.78, 204.68, and 315.88, from the lowest to highest quartiles. In addition, daily ICS dose was associated with NTM occurrence. The incidence rates of NTM disease were 207.63 for individuals receiving low-dose ICS, 261.48 for those receiving medium-dose ICS, and 384.46 for those receiving high-dose ICS. The occurrence rate of NTM according to the ICS types varied. The incidence rate of NTM was 235.17 in fluticasone propionate, 241.31 in budesonide, 249.77 in beclomethasone, 305.88 in ciclesonide, and 240.72 in fluticasone furoate (Table 3).

### 3.3. Risk of NTM in ICS Users

Univariate and multivariate analyses were conducted to compare the risk of NTM occurrence between ICS and non-ICS users (Appendix A and Figure 2). ICS users did not have an increased risk of NTM compared to non-ICS users (HR, 1.121; 95% CI, 0.950–1.323; *p* = 0.176). In a subgroup analysis of ICS users, the highest quartile of cumulative ICS dose was associated with the development of NTM (HR, 1.200; 95% CI, 0.950–1.323; *p* = 0.050). Medium and high daily ICS doses were also associated with an increased risk of NTM disease. The HRs of medium- and high-dose ICS were 1.229 (95% CI, 1.008–1.499; *p* = 0.041) and 1.637 (95% CI, 1.241–2.160; *p* < 0.001), respectively. When analyzing the HR of NTM occurrence by ICS type, no significant differences were identified (all *p* > 0.05): fluticasone propionate, 1.134 (95% CI, 0.937–1.372); budesonide, 1.169 (0.929–1.473); beclomethasone, 1.056 (0.746–1.494); ciclesonide, 1.351 (0.906–2.014); and fluticasone furoate, 0.861 (0.614–1.208).

## 4. Discussion

In this study, we aimed to determine the risk of NTM disease associated with ICS use in a nationwide population-based cohort. Our findings indicated that ICS use might be associated with an increased risk of NTM. The risk of NTM occurrence significantly increased when medium- and high-dose ICS was used. Furthermore, we did not find any difference in the risk of NTM based on the type of ICS used.

NTMs are environmental organisms commonly found in soil, water, and biofilms [17]. They are opportunistic pathogens that cause chronic pulmonary infections. Although the incidence of NTM has increased over the past 30 years, the exact cause has not been identified [18]. Several factors have been proposed as possible causes. First, improved diagnostic technology and increased awareness among healthcare providers have led to the improved detection and reporting of NTM cases [19]. Second, changes in population demographics, such as an aging population and a growing number of individuals with underlying lung conditions, have contributed to an increased susceptibility to NTM disease [20]. Another potential factor related to the increased incidence of NTM is the use of drugs, including corticosteroids, which can increase the risk of compromised immunity.

Andréjak et al. performed the first population-based case-control study using data from the Danish Central Population Registry [21]. Patients with COPD had a 15.7-fold increased risk of developing NTM disease. The odds ratio (OR) for NTM disease was 29.1 (95% CI, 13.3–63.8) in patients with COPD who received current ICS treatment and 7.6 (3.4–16.8) in patients who had never received ICS. Brode et al. performed a population-based nested case-control study using health administrative databases in Ontario, Canada [12]. The study participants were older adults (≥66 years) treated for obstructive lung disease. In individuals with COPD, NTM disease was associated with current ICS (1.96, 1.62–2.36) but not with prior ICS (0.98, 0.79–1.20) use. Another case-control study was conducted by reviewing the medical records of patients registered in the healthcare system of northern California [22]. The study found that the use of ICS was associated with increased odds of NTM disease (2.51, 1.40–4.49). Shu et al. conducted a nested case-control study using a retrospective review of electronic medical records from a tertiary referral medical center in Taiwan and a meta-analysis that included the aforementioned studies [23]. The meta-analysis revealed a 2-fold higher risk of NTM disease in patients who had received ICS within 1 year prior to NTM diagnosis (2.02, 1.41–2.90). However, our study did not show an increased risk of NTM among the total number of ICS users, which is inconsistent with the findings of previous observational studies. Our study design was a retrospective cohort study that included only patients with COPD, while previous case-control studies included patients with chronic obstructive or airway diseases, such as asthma and bronchiectasis. Such differences in study design and subjects could have contributed to these contradictory results.

Previous observational studies have reported a dose–response relationship between ICS dose and the risk of NTM disease. Andréjak et al. reported that the risk of NTM disease increased with an increase in the daily dose of ICS [21]. The OR was 28.1 for low-dose and 47.5 for high-dose ICS. The meta-analysis reported an increasing OR for NTM disease with higher daily doses of ICS: 1.61 (0.91–2.86) for low-dose, 1.85 (1.30–2.86) for medium-dose, and 3.49 (1.92–6.36) for high-dose ICS [23]. Increasing the cumulative ICS dose was also associated with an increased risk of NTM disease [12,22]. These findings are consistent with the results of this study. Although the overall results did not reach statistical significance, there was an association between the highest quartile of cumulative ICS dose and daily medium and high doses of ICS with the development of NTM. Increased risks of pneumonia and tuberculosis with higher daily and cumulative doses of ICS have also been reported in several studies [16,24]. The results of our study strengthen the dose–response relationship between ICS dose and the complications of pulmonary infection in patients with COPD.

Previous studies have reported that the risk of NTM disease differs according to ICS type. A higher OR for NTM disease was identified in patients receiving fluticasone (40.8, 14.0–119.5) compared to those receiving budesonide (19.8, 7.2–54.4) [21]. Another study reported that NTM disease was significantly associated with fluticasone (2.09, 1.80–2.43) but not with budesonide (1.19, 0.97–1.45) [12]. Fluticasone has a more potent immunosuppressive effect than budesonide [25]. Therefore, fluticasone is strongly associated with an increased risk of pneumonia and tuberculosis compared to budesonide [26,27]. However, we did not find a significant difference in the risk of NTM disease based on ICS type. Further studies are warranted to address whether ICS type affects the risk of developing NTM disease.

The strengths of this study are its external validity and improved generalizability as an observational study because we used a nationwide population-based database that encompasses the entire Korean population. Additionally, contrary to other observational studies including chronic obstructive or airway diseases, our study participants consisted of only patients with COPD. However, this study has some limitations. First, a large proportion of ICS users were not included as the final study subjects after matching and 50% sample selection. That might affect the overall study results and statistical power of this study. Second, several factors were significantly different after propensity score matching due to the heterogeneity between ICS and non-ICS users. Different variables were adjusted in the final multivariate analysis. Third, the identification of NTM cases in this study relied on diagnostic codes. Treatment of NTM is recommended in patients who meet all of the clinical, radiological, and microbiological criteria for NTM diagnosis. However, due to the inherent limitations of the claims database, we were unable to differentiate between microbiologically confirmed NTM cases.

## 5. Conclusions

ICS may increase the risk of NTM disease in patients with COPD. Therefore, healthcare providers should individually weigh the potential benefits and risks of ICS use, especially when prescribing high doses and prolonged durations of ICS. Close monitoring and patient education are crucial for the early detection and management of NTM disease. Although some studies have suggested a potential link, the evidence is not conclusive and further research is needed.

## Figures and Tables

**Figure 1 jpm-13-01088-f001:**
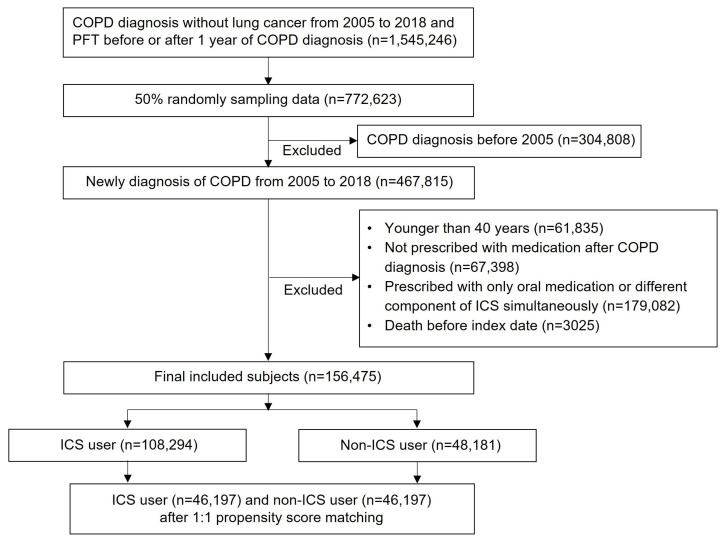
Patient selection flow.

**Figure 2 jpm-13-01088-f002:**
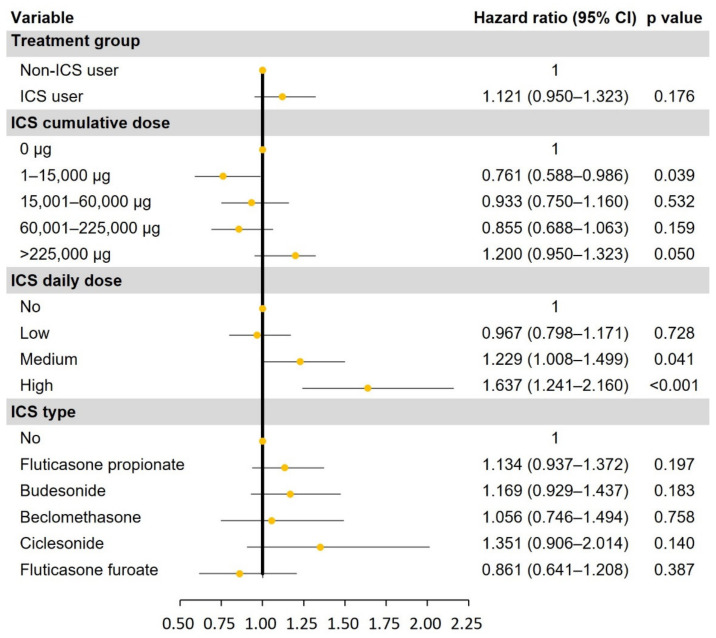
Risk of NTM disease from ICS use.

**Table 1 jpm-13-01088-t001:** Baseline characteristics of study participants.

	ICS Users (*n* = 46,197)		Non-ICS Users (*n* = 46,197)		*p*-Value
	*n*	%	*n*	%	
Age (years)					
Mean (SD)	66.59 (10.94)		66.68 (10.89)		0.186
40–49	3348	7.25	3315	7.18	0.371
50–59	8483	18.36	8602	18.62	
60–69	14,467	31.32	14,661	31.74	
70–79	14,343	31.05	14,141	30.61	
≥80	5556	12.03	5478	11.86	
Sex					
Male	33,641	72.82	33,927	73.44	0.034
Female	12,556	27.18	12,270	26.56	
Comorbidity					
Bronchiectasis	4223	4.68	4307	4.80	0.340
Diabetes	16,028	17.77	15,935	17.76	0.520
Hypertension	28,593	31.70	28,395	31.64	0.180
Heart failure	10,531	11.68	10,497	11.69	0.754
Stroke	11,369	12.60	11,233	12.52	0.298
Chronic kidney disease	3009	3.34	3002	3.35	0.926
Chronic liver disease	16,442	18.23	16,381	18.25	0.675
CCI					
Mean (SD)	3.05 (2.09)		3.10 (2.13)		<0.001
<2	12,644	27.37	12,050	26.08	<0.001
≥2	33,553	72.63	24,147	73.92	
Bronchodilator					
SABA	1593	3.45	2368	38.94	<0.001
LAMA	7261	15.72	18,258	39.52	<0.001
LABA	37,871	68.99	2421	5.24	<0.001
LABA/LAMA	5472	11.84	12,150	26.30	<0.001
Hospitalization before index date					
Yes	2950	6.39	2426	5.25	<0.001
No	43,247	93.61	43,771	94.75	
OCS prescription					
Yes	34,764	75.25	34,416	74.50	0.008
No	11,433	24.75	11,781	25.50	
OCS prescription day					
Mean (SD)	15.64 (26.86)		6.07 (11.87)		<0.001
Interval from COPD diagnosis to index date					
Mean (SD)	378.8 (842.4)		400.9 (866.8)		<0.001

SABA, short-acting agonist; LAMA, long-acting muscarinic antagonist; LABA, long-acting β_2_ agonist; OCS, oral corticosteroid.

**Table 2 jpm-13-01088-t002:** Characteristics of inhaled corticosteroid users.

	ICS Use (*n* = 46,197)	
	*n*	%
ICS cumulative dose (μg)		
Mean (SD)	242,280.8 (616,995.24)	
Median (Q1, Q3)	60,000 (15,000, 225,000)	
0–5000	11,751	25.77
15,001–60,000	11,749	25.77
60,001–225,000	10,814	23.72
>225,000	11,277	24.74
ICS daily dose		
Low	26,919	59.04
Medium	15,202	33.34
High	3470	7.61
Type of ICS		
Fluticasone propionate	21,698	46.97
Budesonide	10,201	22.08
Beclomethasone	4714	10.26
Ciclesonide	1889	4.09
Fluticasone furoate	7668	16.60

**Table 3 jpm-13-01088-t003:** Incidence rate of nontuberculous mycobacterial pulmonary disease according to ICS type and dose.

	Person Year	NTM Patients	Incidence Rate (Per 100,000)
Non-ICS users	192,202.11	440	228.93
ICS users	205,324.15	495	241.08
ICS cumulative dose (μg)			
0–15,000	35,558.75	67	188.42
15,001–60,000	45,879.23	99	215.78
60,001–225,000	48,856.66	100	204.68
>225,000	71,228.87	225	315.88
ICS daily dose			
Low	105,955.62	220	207.63
Medium	78,401.15	205	261.48
High	17,166.75	66	384.46
Types of ICS			
Fluticasone propionate	116,934.20	275	235.17
Budesonide	47,242.98	114	241.31
Beclomethasone	15,614.55	39	249.77
Ciclesonide	8499.94	26	305.88
Fluticasone furoate	17,032.48	41	240.72

## Data Availability

Data were obtained from the National Health Insurance Sharing Service (NHISS) and are available at https://nhiss.nhis.or.kr (accessed 12 April 2023). The NHISS allows access to all of these data for any researcher who promises to follow research ethics at some cost. Those seeking access to this article’s data can download it from the website after promising to follow the research ethics.

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
