# Peer review of "Inhaled Corticosteroids and the Risk of Nontuberculous Mycobacterial Pulmonary Disease in Chronic Obstructive Pulmonary Disease: Findings from a Nationwide Population-Based Study"

_jpm, 2023, doi:10.3390/jpm13071088_

Round 1
Reviewer 1 Report
Assessing potential risks of ICS therapy in COPD is an important endeavor and the large population database is a good cohort for study. The major concerns are threefold.
1. The failure to include a large portion of available ICS users due to matching and the 50% sample selection.
2. Use of OCS for apparently only 1 year in the propensity matching. It would seem to be more appropriate to use total OCS exposure over the study period.
3. The conclusion that ICS is not associated with NTM when the results are statistically significant for those with medium and high dose ICS on a "daily" basis which is the way most ICS is prescribed in COPD.
Another less major concern is the statement that you have only COPD because the participants had a spirometry within 12 months before or after diagnosis. Do you know that these results were consistent with COPD--they have not always been in other studies--again this is minor.
Specific comments--line 87--what % of the available population met the spirometry requirement?
Line 11-113--it appears that OCS and hospitalizations were only assessed for 1 year?
Lines 116-125--this is the description of using OCS in propensity matching which limits the time period over which appears to be assessed.
Line 147-148--this is where the loss of ICS users is described due to failure to identify a non-ICS user match.
LIne 180--this line of NTM being associated with daily ICS use seems to get lost and is very important since in most more developed countries ICS is prescribed for use on a daily basis. How daily use was assessed is not described.
LInes 211-214--The conclusions do not adequately reflect the increased risk of NTM in daily ICS use and with medium and high dose ICS use. Using averages hides the importance of the results and should not represent your primary conclusion.
Author Response
Thank you for encouraging us to revise our manuscript, which we submitted to The Journal of Personalized Medicine. In the revised manuscript, we have addressed the issues raised by the reviewers in their entirety. Please see our point-by-point responses to each of the reviewers’ comments. New or revised sentences are marked in yellow.
Reviewer’s comments
# reviewer 1:
Assessing potential risks of ICS therapy in COPD is an important endeavor and the large population database is a good cohort for study. The major concerns are threefold.
- The failure to include a large portion of available ICS users due to matching and the 50% sample selection.
Reply 1: Thank you for your review and valuable feedback.
As you mentioned, large proportion of ICS users were not included in final study subjects after matching and 50% sample selection. That might impact overall study results and statistical power of our study. We added that point in study limitations.
Changes to the text: However, this study has some limitations. First, large proportion of ICS users were not included in final study subjects after matching and 50% sample selection. That might affect overall study results and statistical power of this study. First Second, several factors were significantly different after propensity score matching due to the heterogeneity between ICS and non-ICS users. Different variables were adjusted in the final multivariate analysis. Second Third, the identification of NTM cases in this study relied on diagnostic codes. Treatment of NTM is recommended in patients who meet all the clinical, radiological, and microbiological criteria for NTM diagnosis. However, due to the inherent limitations of the claims database, we were unable to differentiate between microbiologically confirmed NTM cases (see line 278-287).
- Use of OCS for apparently only 1 year in the propensity matching. It would seem to be more appropriate to use total OCS exposure over the study period.
Reply 2: There was a mistake in the expression of OCS period. OCS exposure in our study was 1year before the index date rather than 1year following the index date. It would seem appropriate to include total OCS exposure over the study period. However, we decided to include OCS exposure only for 1 year before the index date following reference article: “OCS users were defined by any use in the 1 year before the index date. The cumulative dose and the number of prescriptions for oral corticosteroids in the year before the index date were calculated.”
Changes to the text: We added the reference in the description of OCS exposure period.
Prescription records of oral corticosteroids (OCSs) were collected to determine the proportion of subjects who had received an OCS prescription and to calculate the mean duration of the OCS prescription for a period of one year following before the index date [15] (see line 110-113).
Ref) 15. Brassard, P.; Suissa, S.; Kezouh, A.; Ernst, P. Inhaled corticosteroids and risk of tuberculosis in patients with respiratory diseases. Am J Respir Crit Care Med 2011, 183, 675-678.
- The conclusion that ICS is not associated with NTM when the results are statistically significant for those with medium and high dose ICS on a "daily" basis which is the way most ICS is prescribed in COPD.
Reply 3: We modified conclusion as advised. Our study results are significant when reflecting usual clinical practice where medium dose of ICS is usually prescribed to patients with COPD.
Changes to the text:
Our findings indicated that ICS use was not might be associated with an increased risk of NTM. However The risk of NTM occurrence significantly increased when medium- and high-dose ICS was used. Furthermore, we did not find any difference in the risk of NTM based on the type of ICS used (see line 213-217).
Results indicated that ICS users did not have an increased risk of NTM disease compared to non-ICS users (hazard ratio [HR], 1.121; 95% confidence interval [CI], 0.950-1.323; p = 0.176). However, in a subgroup analysis, the highest quartile of the cumulative ICS dose was associated with the development of NTM (1.200, 0.950-1.323, p = 0.050). Medium (1.229, 1.008-1.499, p = 0.041) and high daily doses of ICS (1.637, 1.241-2.160, p < 0.001) were associated with an increased risk of NTM disease. There was no difference in the risk of NTM according to ICS type. ICS use may not increase the risk of developing NTM disease in patients with COPD. However, Physicians should weigh the potential benefits and risks of ICS, especially when using high doses and prolonged durations (see line 21-29).
However, our study did not show an increased risk of NTM among total number of ICS users, which is inconsistent with the findings of previous observational studies (see line 243-245).
ICS may not increase the risk of NTM disease in patients with COPD. However, high ICS doses can increase the risk of NTM disease. Therefore, healthcare providers should individually weigh the potential benefits and risks of ICS use, especially when prescribing high doses and prolonged durations of ICS. Close monitoring and patient education are crucial for early detection and management of NTM disease. Although some studies have suggested a potential link, the evidence is not conclusive and further research is needed (see line 289-294).
Another less major concern is the statement that you have only COPD because the participants had a spirometry within 12 months before or after diagnosis. Do you know that these results were consistent with COPD--they have not always been in other studies--again this is minor.
Reply: In a population-based study for COPD, COPD case was usually identified using diagnostic code with corresponding medication use. In a patient selection procedure of our study, we tried to select COPD case using diagnostic code with medication use. However, the number of COPD cases and magnitude of data exceeded limitation. Therefore, we added spirometry criteria following previously conducted population-based study of COPD.
Changes to the text: We added the reference in the description of spirometry.
The diagnostic codes were classified according to the International Classification of Diseases 10 (ICD-10). COPD cases were defined according to the corresponding diagnostic codes J42–J44 (except J430), procedure of pulmonary function tests, and prescription of inhaled medications [14] (see line 80-83).
Ref) 14. Cho, K.H.; Kim, Y.S.; Linton, J.A.; Nam, C.M.; Choi, Y.; Park, E.C. Effects of inhaled corticosteroids /long-acting agonists in a single inhaler versus inhaled corticosteroids alone on all-cause mortality, pneumonia, and fracture in chronic obstructive pulmonary disease: A nationwide cohort study 2002-2013. Respir Med 2017, 130, 75-84.
Specific comments--line 87--what % of the available population met the spirometry requirement?
Reply: Unfortunately, specific number of populations who met the spirometry requirement was not offered so that we only described the number of populations who met all inclusion criteria in Figure 1.
Line 11-113--it appears that OCS and hospitalizations were only assessed for 1 year?
Reply: If we wanted to assess and match COPD severity for whole study period, OCS exposure and hospitalizations over the study period rather than 1 year before the index date need to be analyzed. When we searched for previously conducted articles regarding ICS study in COPD, diverse periods of covariate were used. As we replied in comment 2, we decided to include OCS exposure and hospitalizations for 1 year before the index date following reference article.
Lines 116-125--this is the description of using OCS in propensity matching which limits the time period over which appears to be assessed.
Reply: We explained the OCS exposure and period in comment 2.
Line 147-148--this is where the loss of ICS users is described due to failure to identify a non-ICS user match.
Reply: We added the limitation as advised (see comment 1).
LIne 180--this line of NTM being associated with daily ICS use seems to get lost and is very important since in most more developed countries ICS is prescribed for use on a daily basis. How daily use was assessed is not described.
Reply: We described the method assessing daily ICS dose: “The daily dose of ICS was calculated by dividing the total prescribed ICS doses by the duration of the prescription” (see line 122-124).
LInes 211-214--The conclusions do not adequately reflect the increased risk of NTM in daily ICS use and with medium and high dose ICS use. Using averages hides the importance of the results and should not represent your primary conclusion.
Reply: We modified conclusion as advised (see comment 3).
Reviewer 2 Report
This large study of real life data on steroid prescription and NTM occurrence in Korea is well written, clear in its aims and has appropriate methods. They use claims data in the mandatory Koream health system to look at ICS prescription in COPD and whether it links to subsequent diagnosis of COPD. They use a random sampling method of all data to keep the database size managable, and propensity matching to try and account for known (and appropriate) differences between ICS users and non-users - whilst this was able to match reasonably well, some differences remain which are inherent to the populations and not possible to address any other way really well. The presentation of data is good, and the conclusions appropriate based on the results. As the diagnosis of NTM is based on coding there is potential for missed cases, where sputum was submitted and positive but no code assigned, or where overall clinical picture was not consistent with the assigned code, however the authors acknowledge this limitation in their discussion. I had only minor questions/comments
1. Why was the random cut of 50% made before the step about time of COPD diagnosis? This step removed a lot of people anyway, so it seems it might have been better to do the 50% cut later on. Either way it does not require new analysis, i was just curious
2. Could decimal points be limited to 2 in the p values in the table of characteristics - it seems unnecessary to have 4
3. Is any information available on whether treatment was given for NTM or what the outcome may be? Eg death, admissions to hospital after NTM diagnosis. I guess these may be the subject of future analyses
Author Response
Thank you for encouraging us to revise our manuscript, which we submitted to The Journal of Personalized Medicine. In the revised manuscript, we have addressed the issues raised by the reviewers in their entirety. Please see our point-by-point responses to each of the reviewers’ comments. New or revised sentences are marked in yellow.
Reviewer’s comments
# reviewer 2:
This large study of real life data on steroid prescription and NTM occurrence in Korea is well written, clear in its aims and has appropriate methods. They use claims data in the mandatory Koream health system to look at ICS prescription in COPD and whether it links to subsequent diagnosis of COPD. They use a random sampling method of all data to keep the database size managable, and propensity matching to try and account for known (and appropriate) differences between ICS users and non-users - whilst this was able to match reasonably well, some differences remain which are inherent to the populations and not possible to address any other way really well. The presentation of data is good, and the conclusions appropriate based on the results. As the diagnosis of NTM is based on coding there is potential for missed cases, where sputum was submitted and positive but no code assigned, or where overall clinical picture was not consistent with the assigned code, however the authors acknowledge this limitation in their discussion. I had only minor questions/comments
- Why was the random cut of 50% made before the step about time of COPD diagnosis? This step removed a lot of people anyway, so it seems it might have been better to do the 50% cut later on. Either way it does not require new analysis, i was just curious
Reply 1: Thank you for your review and valuable feedback.
We described 50% random sampling in method as “Subsequently, half of the participants were selected using random sampling and offered for research purposes, in accordance with the national policy applied to the usage of public databases” (see line 87-89). We also wanted to receive all population database who met inclusion criteria. However, it is a government policy so that we could not change. As you mentioned, large proportion of population were not included in final study subjects after matching and 50% sample selection. That might impact overall study results and statistical power of our study. We added that point in study limitations.
Changes to the text: However, this study has some limitations. First, large proportion of ICS users were not included in final study subjects after matching and 50% sample selection. That might affect overall study results and statistical power of this study. First Second, several factors were significantly different after propensity score matching due to the heterogeneity between ICS and non-ICS users. Different variables were adjusted in the final multivariate analysis. Second Third, the identification of NTM cases in this study relied on diagnostic codes. Treatment of NTM is recommended in patients who meet all the clinical, radiological, and microbiological criteria for NTM diagnosis. However, due to the inherent limitations of the claims database, we were unable to differentiate between microbiologically confirmed NTM cases (see line 278-287).
- Could decimal points be limited to 2 in the p values in the table of characteristics - it seems unnecessary to have 4
Reply 2: We modified decimal points of the p value from 4 to 3 because 2 decimal points could not sufficiently reflect statistical significance of large populations.
Changes to the text: see manuscript, Table 1, and Figure 2.
- Is any information available on whether treatment was given for NTM or what the outcome may be? Eg death, admissions to hospital after NTM diagnosis. I guess these may be the subject of future analyses.
Reply 3: We totally agree with you. We knew ICS can increase the incidence of pulmonary infections such as pneumonia, tuberculosis, and NTM. However, we did not know exactly how ICS affect outcomes of pulmonary infections. Although, we could not present the outcome data regarding NTM prognosis, that topic definitely deserves further studies.